# Evaluating the Utility of Five Gene Fragments for Genetic Diversity Analyses of *Mytella strigata* Populations

Chenxia Zuo [1,2], Tingting Zhang [2], Chenchen Zhang [2], Daotan Zhao [2], Yi Zhu [1,3], Xiaojie Ma [1,3], Haiyan Wang [1], Peizhen Ma [1,*] and Zhen Zhang [1,3,*]

1    Laboratory of Marine Organism Taxonomy & Phylogeny, Qingdao Key Laboratory of Marine Biodiversity and Conservation, Institute of Oceanology, Chinese Academy of Sciences, Qingdao 266071, China
2    College of Life Sciences, Qingdao University, Qingdao 266000, China
3    University of Chinese Academy of Sciences, Beijing 100049, China
*     Correspondence: mapeizhen@qdio.ac.cn (P.M.); zhangzhen@qdio.ac.cn (Z.Z.)

**Abstract:** *Mytella strigata* (Hanley, 1843) is an invasive mussel species that has rapidly spread in China in recent years. Here, we tested the utility of three mitochondrial gene fragments, *COI*, *12S*, and *16S*, and two nuclear gene fragments, *D1 28S* and *18S-ITS1*, for characterizing the levels of genetic diversity among and within populations using 191 *M. strigata* specimens collected in China to aid ongoing efforts to identify the origin of the invasion as well as molecular genetic studies. *M. strigata* exhibited two sex-associated haplogroups according to the *COI* and *12S* sequences. The ratio of female-lineage to male-lineage *COI* and *12S* sequences was 149:22 and 72:7, and the genetic distances between haplogroups were 6.56 and 9.17, respectively. Only one haplotype was detected among the *18S-ITS1* sequences (413 bp), and three haplotypes were detected among the *D1 28S* sequences (296 bp). The haplotype diversity of both the female-lineage *COI* and *12S* sequences was greater than 0.5, and the nucleotide diversity of the *12S*, *16S*, *D1 28S*, and *18S-ITS1* sequences was less than 0.005 in all six populations in China. Our findings indicated that *COI* is the most useful gene fragment for genetic diversity studies of *M. strigata* populations; *D1 28S* and *18S-ITS1* sequences would be useful for species identification because of their low intraspecific diversity. Our genetic analysis of the *COI* sequences revealed Colombia as the most likely origin of *M. strigata* in China and showed that the invasive populations in China have recently experienced or are currently experiencing a population bottleneck.

**Keywords:** adaptive evolution; mitochondrial cytochrome oxidase subunit I; biological invasion; mussel; molecular marker

## 1. Introduction

Biological invasions pose a major threat to native biodiversity and ecosystem health [1], as they have been shown to lead to the decline or extirpation of native species [2], have deleterious effects on ecosystem functions, and affect the global environment [3–5]. The four stages of biological invasions include introduction, establishment, spread, and impact; the eradication of invasive species is often considered a futile endeavor once the establishment stage is reached [6]. Human activities are responsible for the introduction of many marine invasive species, especially through ballast water [7,8]; however, ocean and coastal currents can promote the spread of invasive species over small areas and, thus, increase the complexity of the dynamics of invasions [9]. Population genetic studies are essential for evaluating the current status of invasive populations as well as monitoring and controlling their spread; such studies can also help identify the origins of invasions and provide insights into patterns of genetic variation and molecular evolution [10].

*Mytella strigata* (Hanley, 1843) (=*Mytella charruana* (d'Orbigny, 1846)) is native to the Pacific and Atlantic coasts of tropical America [11]. This species has received increased

research attention because it has been reported in various regions, including Florida [12] and the Indo-West Pacific [13], namely, the Philippines [14], Singapore [11], Thailand [15], India [16], and China [17]. *M. strigata* has spread rapidly in China in recent years, and the invasion of this species is currently thought to be in the establishment and spread stages [18]. Given that there is extensive variation in shell color and shell surface pattern in *M. strigata* and related species [11], mitochondrial cytochrome oxidase subunit I (*COI*) and large subunit of nuclear ribosomal RNA (*28S*) sequences have been used to distinguish *M. strigata* from similar species [11,17,19]. *COI* sequences also have been used to characterize the distribution of haplotypes [13,19]. However, two types of heteroplasmy have been detected in the *COI* sequences of *M. strigata*. The first is characterized by the presence of two distinct sex-associated mitochondrial lineages [20], which is common in bivalves with both egg-transmitted (female-lineage mitochondrial DNA (mtDNA), F-mtDNA) and sperm-transmitted mitochondrial genomes (male-lineage mtDNA, M-mtDNA) [20,21]; this is referred to as the doubly uniparental inheritance (DUI) pattern [22]. The second is characterized by the presence of two different haplogroups in both sexes; only one of these haplogroups has been detected in Brazil [13]. The *p*-distance between two sex-associated *COI* lineages of *M. strigata* ranges from 20.5% to 20.8% [23]. This has limited the use of *COI* sequences for population genetic studies; there is thus a pressing need to identify mitochondrial and nuclear markers that could be used to identify the origin of *M. strigata* invasions and characterize the patterns of molecular variation in *M. strigata*.

Here, we evaluated the utility of five genetic markers, including three mitochondrial gene fragments, *COI*, the small subunit ribosomal RNA (*12S*), and the large subunit ribosomal RNA (*16S*), and two nuclear gene fragments, the D1 region of 28S (*D1 28S*) and a fragment from the small subunit ribosomal RNA to internal transcribed spacer-1 (*18S-ITS1*), for studies of genetic diversity among and within *M. strigata* populations. Population genetic analyses were conducted to characterize genetic variation among Chinese, Colombian, Ecuadorian, and American populations of *M. strigata*. The aims of this study were to (1) detect the intraspecific diversity for the five gene fragments, (2) identify the gene fragments most useful for population genetic analyses, and (3) evaluate the relationships between invasive and native populations to identify the origin of *M. strigata* invasions in China.

## 2. Materials and Methods

A total of 191 *M. strigata* individuals were collected from four provinces in China from November 2020 to August 2022, and these samples were obtained from six populations: Jimei (JM), Shanwei (SW), Xuwen (XW), Zhanjiang (ZJ), Beihai (BH), and Hainan (HN) (Figure 1, Table 1). All samples were identified as *M. strigata* according to the morphological descriptions of Lim et al. (2018), Sanpanich and Wells (2019), and Huang et al. (2021); the main characteristics used to distinguish *M. strigata* from similar species include the subterminal umbos, external shell color, coloration patterns, and 3 or 4 (as many as 7) teeth in the anterior ventral region of the valves. These specimens were preserved in 95% alcohol. A Tiangen DNA kit (DP324, Tiangen Biotech (Beijing) Co., Ltd., Beijing, China) was used to extract total DNA from the adductor muscles per the manufacturer's instructions. The primers used to amplify the *COI*, *12S*, *16S*, *D1 28S*, and *18S-ITS1* sequences are shown in Table 2. Each PCR reaction mixture contained 0.5 μL of each primer, 1.0 μL of DNA, 12.5 μL of 2× Taq PCR MasterMix (PC1120; Beijing Solarbio Science & Technology Co., Ltd., Beijing, China), and 10.5 μL of ddH$_2$O in a total volume of 25 μL. For the *COI*, *12S*, and *16S* sequences, the thermal cycling conditions were as follows: initial denaturation at 95 °C for 3 min; 32 cycles of 95 °C for 30 s, 48 °C for 1 min, and 72 °C for 1 min; and a final extension at 72 °C for 5 min. The thermal cycling conditions for the *D1 28S* and *18S-ITS1* sequences were based on those described in Lim et al. (2018) (Table 2). Agarose gel electrophoresis was used to confirm the PCR products, and an Applied Biosystems 3730xl Genetic Analyzer (Tsingke Biotechnology Co., Ltd., Beijing, China) was used for sequencing.

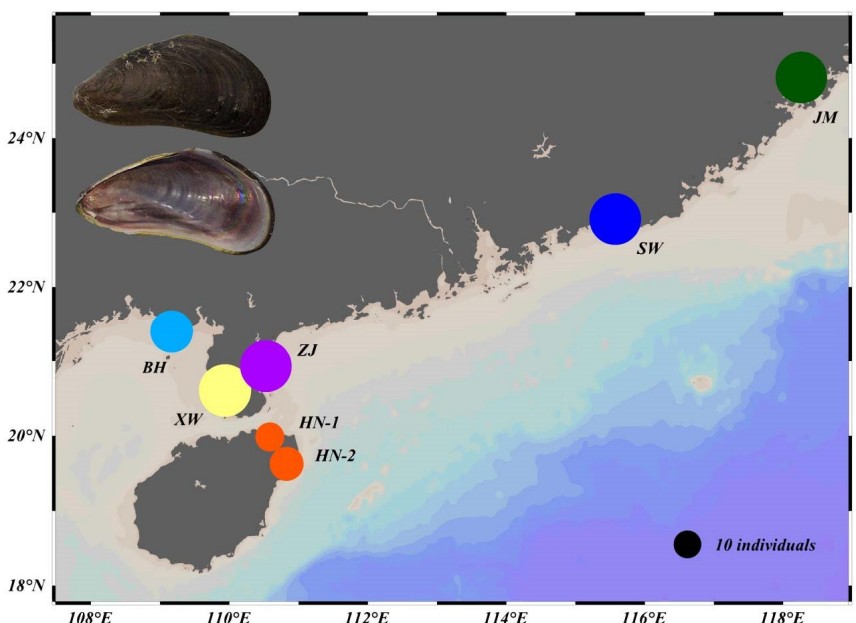

**Figure 1.** Chinese populations of *Mytella strigata* sampled in this study. The size of the circles indicates the number of samples from each population; the black circle (for reference) indicates 10 individuals.

**Table 1.** Information on the samples collected and the number of sequences obtained in this study. The number of male-lineage gene sequences is shown in parentheses.

| Location | Collection Date | Geographical Coordinates | Quantity | *COI* | *12S* | *16S* | *D1 28S* | *18S-ITS1* |
|---|---|---|---|---|---|---|---|---|
| Jimei | 21 October 2021 | 24°34′891″ N, 118°6′578″ E | 35 | 33(1) | 27(1) | 32 | 35 | 35 |
| Shanwei | 31 October 2021 | 22°49′354″ N, 115°33′91″ E | 35 | 29(3) | 25(2) | 33 | 35 | 35 |
| Xuwen | 17 August 2022 | 20°24′739″ N, 109°56′15″ E | 36 | 36(6) | 35(5) | 34 | 36 | 36 |
| Beihai | 12 October 2021 | 21°24′6″ N, 109°10′9″ E | 24 | 20(3) | 21(2) | 21 | 24 | 24 |
| Zhanjiang | 27 November 2020 | 20°55′53″ N, 110°31′43″ E | 35 | 30(1) | 31(1) | 34 | 35 | 35 |
| Hainan | 6 December 2020 | 19°59′7″ N, 110°35′7″ E | 26 | 23(8) | 19(3) | 23 | 26 | 26 |
| | 6 December 2020 | 19°37′35″ N, 110°49′43″ E | | | | | | |
| Total | - | - | 191 | 171(22) | 158(14) | 177 | 191 | 191 |

**Table 2.** Primers and thermal cycling conditions for the PCR reactions of the *COI*, *12S*, *16S*, *D1 28S*, and *18S-ITS1* sequences in this study.

| Markers | Primers | PCR Programs |
|---|---|---|
| *COI* | LCO 1490 5′-GGTCAACAAATCATAAAGATATTGG-3′<br>HCO 2198 5′-TAAACTTCAGGGTGACCAAAAAATCA-3′ [24] | 95 °C 3 min, 32× (95 °C 30 s, 48 °C 1 min, 72 °C 1 min), 72 °C 5 min |
| *12S* | 12S-SR-J14197 5′-GTACAYCTACTATGTTACGACTT-3′<br>12S-SR-N14745 5′-GTGCCAGCAGYYGCGGTTANAC-3′ [25] | 95 °C 3 min, 32× (95 °C 30 s, 48 °C 1 min, 72 °C 1 min), 72 °C 5 min |
| *16S* | 16Sar 5′-CGCCTGTTTATCAAAAACAT-3′<br>16Sbr 5′-CCGGTCTGAACTCAGATCACGT-3′ [26] | 95 °C 3 min, 32× (95 °C 30 s, 48 °C 1 min, 72 °C 1 min), 72 °C 5 min |
| *D1 28S* | LSU5b 5′-ACCCGCTGAAYTTAAGCA-3′<br>D1R 5′-AACTCTCTCMTTCARAGTTC-3′ [11] | 95 °C 5 min, 49 °C 45 s, 72 °C 1 min, 34× (95 °C 30 s, 52 °C 45 s 72 °C 1 min), 95 °C 30 s, 52 °C 45 s, 72 °C 5 min |

**Table 2.** *Cont.*

| Markers | Primers | PCR Programs |
|---|---|---|
| *18S-ITS1* | ITS1A-sal 5′-AAAAAGCTTTTGTACACACCGCCCGTCGC-3′<br>ITS1B-sal 5′-AGCTTGCTGCGTTCTTCATCGA-3′ [11] | 95 °C 3 min, 35× (95 °C 1 min, 45.5 °C 1 min,<br>72 °C 1.5 min), 72 °C 7 min |

Because of sex-related heteroplasmy in *COI*, MEGA 11.0.13 software [27] was used to align all *COI* sequences in Chinese populations with confirmed female-lineage *COI* sequences (*F-COI*, GenBank accession Nos. JQ685156; MG736074; and MG736082) and male-lineage *COI* sequences (*M-COI*, Nos. JQ685158; JQ685159; and MG736069) to identify the lineage corresponding to each sequence [11,23]. Alignment of all *12S* sequences from China in this study revealed two genetically distinct haplogroups: one covering the *12S* sequence that was confirmed from a female (No. MT991018), which was defined as *F-12S*, and the other was identified as male-lineage *12S* (*M-12S*) according to the DUI pattern. EMBOSS Needle (https://www.ebi.ac.uk/Tools/psa/emboss_needle/ accessed on 10 October 2022) was used for the pairwise sequence alignment of both haplogroups of the *COI* and *12S* sequences, and MEGA 11.0.13 software was used to calculate the genetic distance between haplogroups using the maximum composite likelihood method. The ClustalW algorithm in the MEGA 11.0.13 software was used to align the *F-COI*, *F-12S*, *16S*, *D1 28S*, and *18S-ITS1* sequences of Chinese populations. DnaSP 6 [28] software was used to analyze the number of haplotypes (*h*), number of polymorphic (segregating) sites (*S*), haplotype (gene) diversity (*Hd*), nucleotide diversity (*Pi*), and average number of nucleotide differences (*K*) for the five markers in the six populations sampled. MEGA 11.0.13 software was then used to evaluate the inter- and intra-population genetic distances of the *F-COI* and *F-12S* sequences. popART 1.7 [29] and WinArl35 [30] software were used to construct haplotype networks to clarify the patterns of within-species genetic diversity and the genealogical relationships among haplotypes. We also analyzed genetic data from United States, Ecuadorian, and Colombian populations [19].

## 3. Results

A total of 171 *COI* (GenBank accession Nos. OP921310-OP921333), 158 *12S* (GenBank accession Nos. OP935689-OP935698), 177 *16S* (GenBank accession Nos. OP925864-OP925874), 191 *D1 28S* (GenBank accession Nos. OP925881-OP925883), and 191 *18S-ITS1* (GenBank accession No. OP936019) sequences were obtained (Table 1), and the lengths of these sequences were 610, 504, 411, 296, and 413 bp, respectively. Alignment of our sequences with those of *M. strigata* in GenBank revealed two sex-associated haplogroups for both the *COI* and *12S* sequences. The ratio of the F and M haplotypes was 4:20 in the *COI* sequences and 7:3 in the *12S* sequences. The *F-COI* and *F-12S* sequences showed higher identities (identities of 99% and 98%, respectively) than the *M-COI* and *M-12S* sequences (identities of 79% and 76%, respectively).

### 3.1. Genetic Diversity Analyses

The utility of the five markers for analyses of the genetic diversity of *M. strigata* populations varied (Table 3). Two nuclear gene fragments, *D1 28S* and *18S-ITS1*, were highly conserved, and only three haplotypes of *D1 28S* and one haplotype of *18S-ITS1* were detected. Two polymorphic sites were detected in *D1 28S*, and the values of both *Hd* and *K* for *D1 28S* were less than 0.5 (0.490 and 0.496, respectively). The genetic diversity of the three mitochondrial genes was higher than those of the *D1 28S* and *18S-ITS1* sequences, especially the *F-COI* sequences; a total of 22 haplotypes were identified among the *F-COI* sequences, and a total of 8 and 10 haplotypes were identified among the *F-12S* and *16S* sequences, respectively. The values of *Hd*, *Pi*, and *K* for *F-COI* were significantly higher than those for *F-12S* and *16S* ($p < 0.01$). The *Hd* values of both the *F-COI* and *F-12S* sequences were greater than 0.5 in all populations sampled; the *Pi* values for both *F-12S* and *16S* were less than 0.005.

**Table 3.** Genetic diversity parameters for six populations of *Mytella strigata* in China based on five different molecular markers. JM, Jimei; SW, Shanwei; XW, Xuwen; BH, Beihai; ZJ, Zhanjiang; HN, Hainan; *h*, number of haplotypes; *S*, number of polymorphic (segregating) sites; *Hd*, haplotype (gene) diversity; *Pi*, nucleotide diversity; *K*, average number of nucleotide differences.

| | *F-COI* | | | | | *F-12S* | | | | | *16S* | | | | | *D1 28S* | | | | | *18S-ITS1* | | | | |
|---|---|---|---|---|---|---|---|---|---|---|---|---|---|---|---|---|---|---|---|---|---|---|---|---|---|
| | *h* | *S* | *Hd* | *Pi* | *K* | *h* | *S* | *Hd* | *Pi* | *K* | *h* | *S* | *Hd* | *Pi* | *K* | *h* | *S* | *Hd* | *Pi* | *K* | *h* | *S* | *Hd* | *Pi* | *K* |
| JM | 10 | 12 | 0.847 | 0.00454 | 2.756 | 4 | 4 | 0.665 | 0.00303 | 1.526 | 3 | 2 | 0.446 | 0.00115 | 0.474 | 2 | 1 | 0.420 | 0.00142 | 0.420 | 1 | 0 | 0.000 | 0.00000 | 0.000 |
| SW | 10 | 13 | 0.871 | 0.00547 | 3.323 | 7 | 8 | 0.791 | 0.00315 | 1.589 | 6 | 5 | 0.606 | 0.00172 | 0.708 | 2 | 1 | 0.514 | 0.00174 | 0.514 | 1 | 0 | 0.000 | 0.00000 | 0.000 |
| XW | 10 | 14 | 0.832 | 0.00455 | 2.761 | 3 | 3 | 0.600 | 0.00259 | 1.303 | 5 | 4 | 0.439 | 0.00116 | 0.476 | 2 | 1 | 0.464 | 0.00157 | 0.464 | 1 | 0 | 0.000 | 0.00000 | 0.000 |
| BH | 6 | 8 | 0.654 | 0.00344 | 2.088 | 3 | 3 | 0.602 | 0.00261 | 1.310 | 3 | 2 | 0.452 | 0.00116 | 0.476 | 2 | 1 | 0.464 | 0.00157 | 0.464 | 1 | 0 | 0.000 | 0.00000 | 0.000 |
| ZJ | 8 | 11 | 0.766 | 0.00489 | 2.966 | 3 | 3 | 0.625 | 0.00252 | 1.269 | 7 | 9 | 0.629 | 0.00226 | 0.927 | 3 | 2 | 0.528 | 0.00190 | 0.561 | 1 | 0 | 0.000 | 0.00000 | 0.000 |
| HN | 8 | 11 | 0.790 | 0.00559 | 3.390 | 3 | 3 | 0.592 | 0.00154 | 0.775 | 6 | 8 | 0.696 | 0.00294 | 1.209 | 2 | 1 | 0.508 | 0.00172 | 0.508 | 1 | 0 | 0.000 | 0.00000 | 0.000 |
| Total | 22 | 25 | 0.844 | 0.00491 | 2.981 | 8 | 9 | 0.674 | 0.00284 | 1.432 | 10 | 13 | 0.557 | 0.00172 | 0.790 | 3 | 2 | 0.490 | 0.00168 | 0.496 | 1 | 0 | 0.000 | 0.00000 | 0.000 |

## 3.2. Genetic Distance Analyses

Genetic distances between the five populations sampled were similar according to the *F-12S* sequences and ranged from 0.00239 (SW-HN) to 0.00333 (SW-BH) (Table 4). The highest genetic distance within populations was observed in SW (0.00317), and the lowest genetic distance within populations was observed in HN (0.00154). The estimates of the genetic distances between populations were markedly higher according to the *F-COI* sequences and ranged from 0.00429 (JM-BH) to 0.01763 (HN-Ecuador) (Table 5). All the genetic distances between populations from China and from Ecuador or the United States were greater than 0.01, with the exception of the genetic distance between BH and the United States (0.00970). The highest and lowest genetic distance within populations according to the *F-COI* sequences was observed in the United States (0.00880) and BH (0.00369), respectively. The genetic distances between populations from China and Colombia were similar to those between the six Chinese populations.

**Table 4.** Genetic distances between populations (below the diagonal) and genetic distances within six Chinese populations (in bold values) of *Mytella strigata* according to the partial *F-12S* gene. JM, Jimei; SW, Shanwei; XW, Xuwen; BH, Beihai; ZJ, Zhanjiang; HN, Hainan.

| | **SW** | **ZJ** | **JM** | **HN** | **BH** | **XW** |
|---|---|---|---|---|---|---|
| SW | **0.00317** | | | | | |
| ZJ | 0.00307 | **0.00253** | | | | |
| JM | 0.00333 | 0.00273 | **0.00305** | | | |
| HN | 0.00239 | 0.00278 | 0.00301 | **0.00154** | | |
| BH | 0.00334 | 0.00251 | 0.00276 | 0.00318 | **0.00261** | |
| XW | 0.00332 | 0.00252 | 0.00276 | 0.00314 | 0.00250 | **0.00260** |

**Table 5.** Genetic distances between populations (below the diagonal) and genetic distances within six Chinese populations (in bold values) and three western hemisphere populations (United States, Ecuador, and Colombia) of *Mytella strigata* according to variation of the partial *F-COI* gene. JM, Jimei; SW, Shanwei; XW, Xuwen; BH, Beihai; ZJ, Zhanjiang; HN, Hainan. Populations from Ecuador and Colombia are natural populations, and the United States population is an invasive population.

| | **SW** | **ZJ** | **JM** | **HN** | **BH** | **XW** | **Ecuador** | **Colombia** | **USA** |
|---|---|---|---|---|---|---|---|---|---|
| SW | **0.00587** | | | | | | | | |
| ZJ | 0.00556 | **0.00523** | | | | | | | |
| JM | 0.00559 | 0.00525 | **0.00486** | | | | | | |
| HN | 0.00604 | 0.00575 | 0.00652 | **0.00577** | | | | | |
| BH | 0.00492 | 0.00454 | 0.00429 | 0.00576 | **0.00369** | | | | |
| XW | 0.00537 | 0.00501 | 0.00478 | 0.00613 | 0.00409 | **0.00476** | | | |
| Ecuador | 0.01686 | 0.01700 | 0.01666 | 0.01763 | 0.01588 | 0.01649 | **0.00522** | | |
| Colombia | 0.00561 | 0.00538 | 0.00498 | 0.00644 | 0.00444 | 0.00495 | 0.01624 | **0.00515** | |
| USA | 0.01069 | 0.01084 | 0.01044 | 0.01146 | 0.00970 | 0.01037 | 0.01399 | 0.01008 | **0.00880** |



### 3.3. Genetic Relationships of Haplotypes

Two sex-associated haplogroups were identified in the TCS Network constructed using the *12S* sequences (Figure 2), and the genetic distance between these two haplogroups was 9.17. The female-lineage haplogroup was more common than the male-lineage haplogroup both in terms of haplotype quantity (7:3) and sequence quantity (72:7). The most common three haplotypes were Hap_A (61 sequences), Hap_C (47 sequences), and Hap_D (32 sequences). Two sex-associated haplogroups were also identified in the TCS Network constructed using all 171 *COI* sequences from China and 155 *COI* sequences from the United States, Ecuador, and Colombia (Figure 3), and the genetic distance between these two haplogroups was 6.56. The ratio of the *F-COI* sequences to the *M-COI* sequences was 149:22. Only two *COI* haplotypes were shared among populations in the United States and China; no shared haplotypes were detected between Ecuador and China. Ten Colombian haplotypes were detected in China, accounting for five of the six Colombian haplotypes and one of the two F-type haplotypes from China. A total of nine *F-COI* haplotypes and four *M-COI* haplotypes from China were not detected in the United States, Ecuador, or Colombia. The four dominant haplotypes, Hap_1, Hap_2, Hap_4, and Hap_7, were detected in all six Chinese populations.

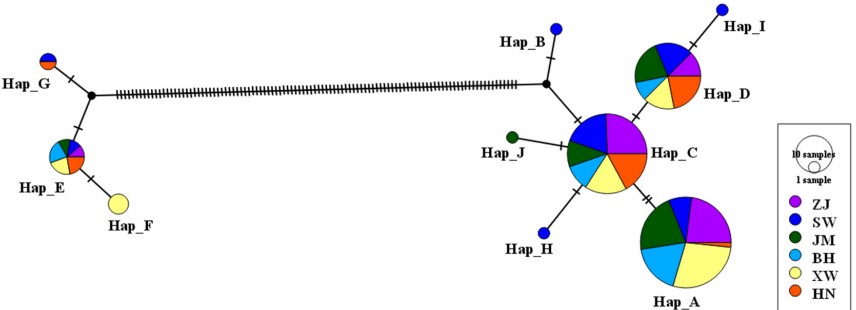

**Figure 2.** A TCS network showing the genetic relationships for the mitochondrial *12S* gene fragment among six Chinese populations of *Mytella strigata*. The male mitochondrial lineage is on the left and the female one is on the right.

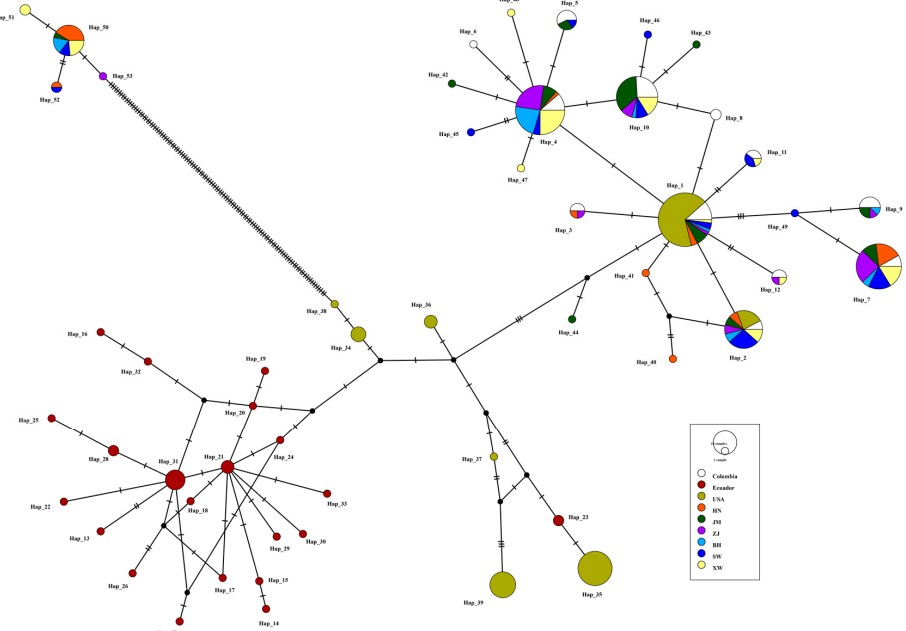

**Figure 3.** A TCS network showing the genetic relationships for the mitochondrial *COI* gene fragment among nine populations of *Mytella strigata* in China, the United States, Colombia, and Ecuador. The cluster of the male lineage of Hap_50–53 is on the upper left, and the other haplotypes are of female lineage.

## 4. Discussion

Genetic diversity indicates the total genetic variation among individuals within and between populations [31,32], and analyses of spatial and temporal variations in the genetic diversity of invasive species are frequently conducted [33,34]. Seasons can affect the genetic diversity of populations, and a high genetic heterogeneity always appears in summer populations [35]; thus, we sampled our populations during the autumn and winter months. The nuclear *D1 28S* and *18S-ITS1* sequences were the most conserved among the five molecular markers examined in this study; these sequences would therefore be useful for distinguishing *M. strigata* from other similar species [23]. The rate of evolution of mtDNA is thought to be faster in animals than that of nuclear DNA [36,37], and the levels of genetic diversity observed in the three mitochondrial genes and two nuclear genes of *M. strigata* provide support for this general finding.

We detected two highly divergent haplogroups in the *COI* and *12S* sequences of *M. strigata*; however, this was not the case for the *16S* sequences. The number of F-type haplotypes was much greater than that of M-type haplotypes, which suggests that the F genome evolves faster than the M genome [38]. The heteroplasmy in *COI* is thought to stem from the DUI pattern [19]. However, all DNA samples were obtained from adductor muscles in this study; 22 of the 171 *COI* sequences were *M-COI*, which is not consistent with the canonical DUI pattern in which all *COI* sequences would be *F-COI* sequences [20]. There are three possible, non-mutually exclusive explanations for this finding: (1) the introgression of specific male-lineage genes, such as *M-COI* and *M-12S*, in *M. strigata*, which results in an unusual DUI pattern [23]; (2) mitochondrial heteroplasmy in which both mitogenomes are present in many tissues of both sexes [39]; and (3) variation in the efficacy of the same pairs of primers for amplifying two lineages sequences [40]. In species with unusual DUI patterns, the presence of M-mtDNA in the adductor muscles of female individuals might reflect sperm mitochondria not having been eliminated in the eggs and that it had dispersed randomly in the blastomeres; in male individuals, sperm mitochondria might have broken away, aggregated, and migrated to adjacent adductor muscle cells [41–43]. The percentage of nucleotide divergence (*p*-distance) between the two types of *COI* sequences varied among species showing a DUI pattern; for example, the *p*-distance between two types of the *COI* sequences was 8% in the veneroid *Artica islandica* [44], 17% in the nuculanoid *Ledella sublevis* [45], 20.5% to 20.8% in *M. strigata* [23], 24% in the mytiloid *Mytilus edulis* [46], and 50% in the unionoid *Inversidens japanensis* [47]. No DUI pattern has been detected in the Mediterranean alien species *Brachidontes pharaonic* despite mitochondrial heteroplasmy (*p*-distance of 8.6%) [39]; this suggests that the DUI pattern might be absent in *M. strigata*. The *16S* sequences were relatively conserved in *M. strigata*; however, variation in the *16S* sequences has been detected in *Donax vittatus* and other species in which a DUI pattern has been detected, including *Mytilus galloprovincialis* [48] and *Perumytilus purpuratus* [49]. Variation in the *12S* sequences was observed in this study, and this is consistent with the results of studies of other mollusk species. Additional mitochondrial data are needed to clarify patterns of genetic differentiation observed among DUI species.

The haplotype diversity of *M. strigata* in China was high (greater than 0.5), and the nucleotide diversity of *M. strigata* in China was low (less than 0.005), indicating that the *M. strigata* population in China has undergone a population bottleneck, followed by rapid population growth and the accumulation of mutations, a pattern that has often been observed in various populations of marine fishes [50]. However, the *16S* gene in *M. strigata* was the most highly conserved among the three mitochondrial markers, and this was reflected by its low haplotype diversity, nucleotide diversity, and average number of nucleotide differences in these sequences. Consequently, the haplotype diversity values of the *16S* sequences in JM, XW, and BH (less than 0.5) do not provide an accurate reflection of the actual genetic status of *M. strigata* based on our analysis of all populations in which all molecular markers were used. Although our genetic distance and haplotype network analyses revealed low levels of genetic differentiation among Chinese populations regardless of whether *COI* or

*12S* gene fragments were used in these analyses, the performance of the *COI* sequences in these analyses was higher than that of the *12S* sequences. The *12S* sequence was shorter than the *COI* sequence [51], and the *F-12S* sequence in our study was 106 bp shorter than the *F-COI* sequence. All the diversity parameters and genetic distances were higher when they were estimated using the *F-COI* sequences than when they were estimated using the *F-12S* sequences, suggesting that the rate of evolution of the *COI* sequences was greater than that of the *12S* sequences [52] and that the *COI* sequences are the most useful for genetic diversity analyses of *M. strigata* populations among the five markers examined.

The genetic distances between Chinese populations and native Colombian populations were lower than those between Chinese populations and the invasive United States population [19,53], which provides support for the hypothesis of Ma et al. [13]: that the invasive populations of *M. strigata* in China might have been derived from Colombian populations. A star-shaped network was observed for Hap_1, Hap_2, and Hap_10 of the *COI* sequences, which indicates that *M. strigata* populations in China might have undergone recent founder events and that they recently experienced or are currently experiencing a population bottleneck [50].

## 5. Conclusions

In this study, the utility of three mitochondrial gene fragments and two nuclear gene fragments were tested for characterizing the levels of genetic diversity among and within populations of *M. strigata* specimens collected in China. *M. strigata* exhibited two sex-associated haplogroups according to the *COI* and *12S* sequences. Our findings indicated that *COI* is the most useful gene fragment for genetic diversity studies of *M. strigata* populations; *D1 28S* and *18S-ITS1* sequences would be useful for species identification. Our genetic analysis of the *COI* sequences revealed Colombia as the most likely origin of *M. strigata* in China and showed that the invasive populations in China have recently experienced or are currently experiencing a population bottleneck.

**Author Contributions:** C.Z. (Chenxia Zuo) and P.M. were responsible for the investigation, data analysis, and original draft; C.Z. (Chenchen Zhang), D.Z., Y.Z. and X.M. helped collect samples and carry out the data analysis; T.Z. and H.W. reviewed and edited the manuscript; Z.Z. was in charge of the methodology and funding provision. All authors contributed to the article and approved the submitted version. All authors have read and agreed to the published version of the manuscript.

**Funding:** This work was supported by the National Key R&D Program of China (No. 2022YFD2401204), the National Natural Science Foundation of China (No. 42006080), and the Science & Technology Fundamental Resources Investigation Program (No. 2022FY100300).

**Institutional Review Board Statement:** Ethical review and approval were waived for this study, because the invasive mussel in this study is an invertebrate with no sense or subjective experience.

**Informed Consent Statement:** Not applicable.

**Data Availability Statement:** Relevant information has been added in the article.

**Acknowledgments:** We appreciate the help from Liqiang Zhao from Guangdong Ocean University for the sample collection. We would also like to thank the reviewers for their constructive comments and helpful suggestions that improved the manuscript.

**Conflicts of Interest:** The authors declare no conflict of interest.

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
