# Peer review of "Evaluating the Utility of Five Gene Fragments for Genetic Diversity Analyses of Mytella strigata Populations"

_fishes, doi:10.3390/fishes8010034_

Round 1
Reviewer 1 Report
Mytella strigata is an invasive mussel in China, for which we would like to know its genetic diversity, population structure and origin. Zuo et al. evaluated the utility of three mitochondrial and two nuclear regions to characterize variation among 191 specimens collected along the Chinese coast. They observed both female and male mitochondrial haplogroups, which is not surprising given double uniparental inheritance of mitochondria in mollusks. Mitochondrial COI and nuclear ITS1 segments were most useful for the purpose set out, and Columbian seems the most likely source for the invasive mussels in China. The design and implementation of the study are straightforward, and the write-up is mostly fine. I appreciate that the authors utilized an English editing service. I‘ll present comments here needing elaboration and have written suggested fixes on the manuscript document as well.
Abstract. – With a sample size of 191, it is not defensible to present genetic distance parameters to five decimal places, as occurs at line 18 and several passages throughout the document. I’ve marked the text.
At line 26, the mussel likely experienced its genetic bottleneck at the founding event, so it is best to write the sentence as the invasive populations having recently experienced or are currently experiencing a population bottleneck. Indeed, the star-shaped haplotype networks suggest population expansion and retention of new mutations, so the bottleneck is more likely recent that current.
Introduction. – I have written just a few marks on the manuscript document.
Methods. – At line 85, the structure referred to is the umbo, plural umbos.
Supporting citations are needed for all software packages used; these are lacking at lines 119, 121, and 126.
Results. – The label for Table 5 should tell the reader what genetic distance metric is reported in the body of the table. Is this p-distance?
Figure 2 can be presented the full width of the page to make it more legible. The legend is inadequate, as it does not tell the reader that this is for the mitochondrial 12S gene fragment. The label should tell the reader that the male mitochondrial lineage in on the left and the female on the right.
Figure 3 and its label present the same issues as Figure 2. Tell the read that this is for the mitochondrial COI gene and which cluster is for the male lineage and which for the female.
Discussion. – At line 206, genetic diversity may be considered not only within populations as stated, but also among populations or species.
The sentence at line 209 is unsupported or misleading as presented. The authors write that genetic heterogeneity in THEIR population is higher in summer than at other seasons, yet no data are presented to substantiate that claim, and reference [31] is for another species. This must be resolved.
At line 227, it is becoming more widely recognized that the male mitochondrial lineage is expressed not only in testis as originally observed by Zouros et al., but in many tissues at low levels as the authors did in this study.
At line 274, the issue of whether the species recently experienced a genetic bottleneck or is currently undergoing a genetic bottleneck again comes up.
References. – I appreciate how clean the references are. I found only 1 issue. Kunth should be capitalized at line 351.

Reviewer 2 Report
In this manuscript, Zuo and colleagues investigated the within-species genetic diversity and affinity of M. strigate populations (191 individuals) in southern China using five gene fragments (three from mtDNA and two from nuclear DNA). Overall, the paper is well-written and of general interest to M. strigata researchers. The analyses are suitable, and the conclusions are reasonable based on the results discussed.
I have two minor suggestions.
1- it would be great to add a species photo in Figure 1, for readers.
2- Line 26: add “likely” before “currently” to prevent a confirmative conclusion based on a limited number of molecular markers rather than genome-wide genetic markers.
